# Synergy between Rhizobial Co-Microsymbionts Leads to an Increase in the Efficiency of Plant–Microbe Interactions

**DOI:** 10.3390/microorganisms11051206

**Published:** 2023-05-04

**Authors:** Vera Safronova, Anna Sazanova, Andrey Belimov, Polina Guro, Irina Kuznetsova, Denis Karlov, Elizaveta Chirak, Oleg Yuzikhin, Alla Verkhozina, Alexey Afonin, Igor Tikhonovich

**Affiliations:** 1All-Russia Research Institute for Agricultural Microbiology (ARRIAM), Sh. Podbelskogo 3, 196608 St. Petersburg, Russia; 2Siberian Institute of Plant Physiology and Biochemistry (SIPPB), P.O. Box 1243, 664033 Irkutsk, Russia; 3Department of Genetics and Biotechnology, Saint Petersburg State University, Universitetskaya Emb. 7/9, 199034 St. Petersburg, Russia

**Keywords:** plant–microbe interaction, crop legumes, relict symbiotic systems, rhizobial synergy, efficiency of symbiosis

## Abstract

Combined inoculation of legumes with rhizobia and plant growth-promoting rhizobacteria or endophytes is a known technique for increasing the efficiency of nitrogen-fixing symbiosis and plant productivity. The aim of this work was to expand knowledge about the synergistic effects between commercial rhizobia of pasture legumes and root nodule bacteria of relict legume species. Pot experiments were performed on common vetch (*Vicia sativa* L.) and red clover (*Trifolium pratense* L.) co-inoculated with the participation of the corresponding commercial rhizobial strains (*R. leguminosarum* bv. *viciae* RCAM0626 and *R. leguminosarum* bv. *trifolii* RCAM1365) and seven strains isolated from nodules of relict legumes inhabiting the Baikal Lake region and the Altai Republic: *Oxytropis popoviana*, *Astragalus chorinensis*, *O. tragacanthoides* and *Vicia costata.* The inoculation of plants with combinations of strains (commercial strain plus the isolate from relict legume) had a different effect on symbiosis depending on the plant species: the increase in the number of nodules was mainly observed on vetch, whereas increased acetylene reduction activity was evident on clover. It was shown that the relict isolates differ significantly in the set of genes related to different genetic systems that affect plant–microbe interactions. At the same time, they had additional genes that are involved in the formation of symbiosis and determine its effectiveness, but are absent in the used commercial strains: symbiotic genes *fix*, *nif*, *nod*, *noe* and *nol*, as well as genes associated with the hormonal status of the plant and the processes of symbiogenesis (*acdRS*, genes for gibberellins and auxins biosynthesis, genes of T3SS, T4SS and T6SS secretion systems). It can be expected that the accumulation of knowledge about microbial synergy on the example of the joint use of commercial and relict rhizobia will allow in the future the development of methods for the targeted selection of co-microsymbionts to increase the efficiency of agricultural legume–rhizobia systems.

## 1. Introduction

The term rhizobia can be considered as a historically valid general term for a large group of predominantly soil- and plant-associative α-proteobacteria that form the order *Rhizobiales* Kuykendall 2006 (synonym *Hyphomicrobiales* Douglas 1957 emend. Hördt et al. 2020) [1]. Representatives of the first described rhizobia species were isolated from root nodules of leguminous plants (family *Fabaceae*) and belonged to the genera *Allorhizobium*, *Azorhizobium*, *Bradyrhizobium*, *Sinorhizobium*, *Rhizobium* and *Mesorhizobium* [2]. Subsequently, numerous other genera and species within this order were isolated from nodules, although many of the described taxons did not nodulate the host plant upon inoculation (e.g., *Agrobacterium*, *Bosea*, *Tardiphaga*, some *Phyllobacterium* and *Microvirga* species). In addition to rhizobia, several genera of β-proteobacteria (e.g., *Burkholderia*, *Cupriavidus*, *Herbaspirillum*) were included among the legumes nodulating bacteria [3,4,5].

In recent years, there has been increasing interest in non-rhizobial endophytes within root nodules [6,7,8,9]. It was shown that the nodule occupants may belong to taxonomically distant genera: *Paenibacillus* (*Bacillales*); *Inquilinus* and *Paracraurococcus* (*Rhodospirillales*); *Sphingomonas* (*Sphingomonadales*); *Pseudomonas* (*Pseudomonadales*); *Agromyces*, *Ornithinicoccus* and *Microbacterium* (*Micrococcales*), and so on [10]. Among 300 strains isolated from the root nodules of seven legume species in different ecological zones of China, only 13% belonged to the order *Rhizobiales*. Most of other isolates were from the orders *Bacillales* (64%) and *Pseudomonadales* (12%). The remaining isolates belonged to *Burkholderiales*, *Paenibacillales*, *Enterobacteriales*, *Actinomycetales*, *Sphingomonadales*, *Xanthomonadales*, *Chitinophagales*, *Brevibacillales*, *Staphylococcales* and *Mycobacteriales* [11].

It should be noted that the possible roles of bacterial inhabitants of nodules (both rhizobial and non-rhizobial) in the plant–microbe interactions have not yet been sufficiently studied. However, their populations have been found to have multiple plant growth-promoting (PGP) characteristics, including the production of auxins, siderophores, 1-aminocyclopropane-1-carboxylate deaminase and hydrogen cyanide; hydrolytic enzymatic activity (e.g., lipase, cellulase, protease and chitinase); phosphate solubilization; and inhibition of pathogenic fungi [12]. The positive effect of these bacteria on the growth and yield of legumes was confirmed via co-inoculations together with nodule-forming rhizobia strains [13,14,15].

Previously, we suggested that a promising model for studying the natural diversity of bacterial inhabitants of root nodules can be relict plant-microbial systems that are in the process of forming symbiotic relationships. From the nodules of the Miocene-Pliocene relict legumes *Vavilovia formosa*, *Oxytropis triphylla*, *O. popoviana*, *Astragalus chorinensis* and *Caragana jubata*, a wide range of microsymbionts, including representatives of the families *Phyllobacteriaceae*, *Rhizobiaceae*, *Bradyrhizobiaceae*, *Boseaceae* and *Nitrobacteraceae*, have been isolated [16,17,18,19,20,21]. Notably, plant nodules often contained two taxonomically different rhizobial strains. As a rule, one of them was nodule-forming (e.g., *Rhizobium*, *Mesorhizobium* and *Bradyrhizobium*) and the other was non-nodulating (e.g., *Phyllobacterium*, *Bosea* and *Tardiphaga*). It was shown that relict microsymbionts have various combinations of symbiotic genes, as well as genes that can affect rhizobia–legume interactions (production of phytohormones and virulence factors) [19,21]. Using these isolates in plant nodulation assays, the phenomenon of bacterial synergy was discovered, which is expressed in the ability of different strains to localize in the same nodules and increase the parameters of symbiosis (the rate of nodule formation, number of nodules, nitrogen-fixing activity and plant biomass) due to the presence of complementary genes [18,19,21]. These results suggested that rhizobial co-microsymbionts of relict plants are jointly involved in the relationship between partners and can be used to increase the productivity of a wide range of host plants, including agricultural legumes.

In this regard, the purpose of this work was to study in pot experiments the effectiveness of symbiosis upon co-inoculation of *V. sativa* (common vetch) and *T. pratense* (red clover) with their commercial strains and strains isolated from the relict plants *O. popoviana* and *A. chorinensis*, originating from the Baikal Lake region, as well as from *O. tragacanthoides* and *Vicia costata*, growing in the Altai Republic in Russia. The choice of Baikal isolates was carried out on the basis of data from comparative whole genome analysis and plant nodulation assays obtained earlier [21]. These isolates were used here for a comparative study as positive controls. Altai isolates were selected in this study after analyzing the presence of the symbiotic genes *fix*, *nif*, *nod*, *noe* and *nol*, as well as genes that promote plant growth and the formation of symbiosis (*acdRS*; genes associated with the biosynthesis of gibberellins and auxins; genes of T3SS, T4SS and T6SS secretion systems), which complement the commercial strains *Rhizobium leguminosarum* bv. *viciae* RCAM0626 and *R. leguminosarum* bv. *trifolii* RCAM1365.

## 2. Materials and Methods

### 2.1. Bacterial Material

Commercial strains of rhizobia: *Rhizobium leguminosarum* bv. *viciae* RCAM0626 nodulating common vetch (*V. sativa* L.) and *R. leguminosarum* bv. *trifolii* RCAM1365 nodulating red clover (*T. pratense* L.) were used. The rhizobial strains *Mesorhizobium japonicum* Opo-235, *M. japonicum* Opo-242, *Bradyrhizobium* sp. Opo-243 and *M. kowhaii* Ach-343 were isolated previously from nodules of the relict legumes originating from Buryatia (Baikal Lake region, Russia): *Oxytropis popoviana* Peschkova (N51°22′37.67″, E106°28′24.06″) and *Astragalus chorinensis* Bunge (N51°22′35.47″, E106°28′25.12″). The strains *Devosia* sp. A8/3-2, *Phyllobacterium* sp. A18/5-2 and *P. zundukense* A18/3m were isolated in this work from nodules of the relict legumes growing in the Altai Republic (Russia): *O. tragacanthoides* Fisch. ex DC. (N49°45’24.21″, E88°25′52.36″) and *Vicia costata* Ledeb. (N49°39′37.69″, E89°05′24.01″). Information about strains used in this work is given in Table 1. All isolates belong to the order *Rhizobiales* (*Hyphomicrobiales*). Strains of nodule bacteria were isolated from the obtained nodules using the standard method described earlier [22] and were cultivated using modified yeast extract mannitol agar (YMA) [23] supplemented with 0.5% succinate (YMSA) [24]. All strains were deposited in the Russian Collection of Agricultural Microorganisms (RCAM) and stored at −80 °C in an automated Tube Store (Liconic Instruments, Lichtenstein) [25].

### 2.2. Pot Experiments

Seeds of common vetch (*V. sativa*) and red clover (*T. pratense*) plants were surface sterilized with H_2_SO_4_ for 10 min, washed with sterile tap water and germinated on filter paper in Petri dishes at 25 °C in the dark for 4 days. Germinated seedlings were transferred to polypropylene pots containing 0.5 kg of sterile sand (2 seeds per pot, 7 pots per treatment). Each pot was supplemented with 100 mL of the nutrient solution (g/L): K_2_HPO_4_ 1.0, KH_2_PO_4_ 0.5, MgSO_4_ 1.0, Ca_3_(PO_4_)_2_ 0.2, FeSO_4_ 0.02, H_3_BO_3_ 0.005, (NH_4_)_2_MoO_4_ 0.005, ZnSO_4_ × 7 H_2_O 0.005, MnSO_4_ 0.002. Seedlings were inoculated with individual strains or with a pair of strains in the amount of 10^7^ cells of each strain per pot. The uninoculated plants were used as negative control. Plants were cultivated for 28 days in the growth chamber with 50% relative humidity and four-level illumination/temperature modes: night (dark, 18 °C, 8 h), morning (200 μmol m^−2^ s^−1^, 20 °C, 2 h), day (400 μmol m^−2^ s^−1^, 23 °C, 12 h), evening (200 μmol m^−2^ s^−1^, 20 °C, 2 h). Illumination was performed with L 36W/77 FLUORA lamps (Osram, Munich, Germany). At the end of experiment, the nodules were counted and the fresh biomass of plants (shoots, roots and total biomass) was determined. The nitrogen fixation of nodules was measured using the acetylene-reduction method [26]. Briefly, the roots of each pot (two plants) were collected, washed in tap water, weighted and placed in 50 mL flasks. The flasks were hermetically sealed, supplemented with 5 mL of acetylene and incubated for 1 h in the dark at 20 °C. The amount of ethylene formed in the flasks was determined using a gas chromatograph GC-2014 (Shimadzu, Kyoto, Japan). Roots of control plants without nodules were used as a negative control. Then, nodule number in each flask was calculated. Strains were re-isolated from the obtained nodules and identified using 16S rDNA sequencing as described earlier [27]. The data were processed via the standard method of variance analysis using the software STATISTICA version 10 (StatSoft Inc., Tulsa, OK, USA). Fisher’s LSD test was used to evaluate differences between means.

### 2.3. Whole Genome Sequencing of the Strains under Investigation

Genome sequencing of the Baikal isolates *Mesorhizobium japonicum* Opo-235, *M. japonicum* Opo-242, *Bradyrhizobium* sp. Opo-243 and *M. kowhaii* Ach-343 was performed on a MiSeq genomic sequencer (Illumina, San Diego, CA, USA), as described earlier [18,19]. For the whole genome sequencing of two commercial rhizobial strains deposited in the RCAM collection (*R. leguminosarum* bv. *Viciae* RCAM0626 and *R. leguminosarum* bv. *Trifolii* RCAM1365) as well as three Altai isolates (*Devosia* sp. A8/3-2, *Phyllobacterium* sp. A18/5-2 and *P. zundukense* A18/3m), their genomic DNA was extracted using a Genomic DNA Purification KIT (Thermo Fisher Scientific, Waltham, MA, USA) according to the manufacturer’s recommendations. Long-read whole genome sequencing was performed using a MinIon sequencer (Oxford Nanopore, Oxford Science Park, Oxford, UK) of the Core Centrum “Genomic Technologies, Proteomics and Cell Biology” at the ARRIAM. A SQK-LSK109 Ligation Sequencing Kit with an EXP-NBD104 Native Barcoding Expansion 1-12 kit was used to prepare the library according to the manufacturer’s instructions. The reads were base called and demultiplexed using guppy_basecaller (v. 3.3.0). A Flye pipeline (v 2.6-release) was used to assemble the Nanopore reads [28]. The resulting assembly was corrected 4 times using Racon (v1.3.2, https://github.com/lbcb-sci/racon, accessed on 1 March 2020) with the following modifiers (-m 8 -x -6 -g -8 -w 500), followed with a single polish using the medaka (v 0.10.0, https://github.com/nanoporetech/medaka, Oxford Nanopore Technologies, Oxford, UK, accessed on 13 March 2020) program with default parameters [29]. Search for genes in the assembled contigs was performed using the RAST annotation service [30]. All the genomes were assembled into circular replicons. 

Search for homologs of the symbiotic *fix*, *nif*, *nod*, *noe* and *nol* genes as well as genes that promote plant growth (*acdRS*, gibberellin- and auxin-biosynthesis related) and genes of the T3SS, T4SS and T6SS secretion systems in annotated genomes was performed using CLC Genomics Workbench 7.5.1 (CLC bio, Aarhus, Denmark) software using local BLASTn and tBLASTx (e-value threshold of 1e-30) (https://blast.ncbi.nlm.nih.gov accessed on 22 March 2023). 

The whole genome sequences have been deposited to the NCBI GenBank database under accession numbers: CP050553-CP050557 and CP050514-CP050519 for the commercial strains *R. leguminosarum* bv. *viciae* RCAM0626 and *R. leguminosarum* bv. *trifolii* RCAM1365, respectively; QKOD00000000, MZXX00000000, MZXW00000000 and MZXV00000000 for the Baikal isolates *M. japonicum* Opo-235, *M. japonicum* Opo-242, *Bradyrhizobium* sp. Opo-243 and *M. kowhaii* Ach-343, respectively; CP104974-CP104975, CP104966-CP104968 and CP104969-CP104973 for the Altai isolates *Devosia* sp. A8/3-2, *Phyllobacterium* sp. A18/5-2 and *P. zundukense* A18/3m, respectively.

## 3. Results and Discussion

### 3.1. Pot Experiments

The pot experiments were performed on common vetch and red clover plants. Commercial strains of rhizobia (*R. leguminosarum* bv. *viciae* RCAM0626 and *R. leguminosarum* bv. *trifolii* RCAM1365), as well as seven strains isolated from the relict legumes of the Baikal Lake region and the Altai Republic (*Oxytropis popoviana*, *Astragalus chorinensis*, *O. tragacanthoides* and *Vicia costata*), were used for mono- and co-inoculation (Table 1). It should be noted that the climate in the places of growth of relict legumes is sharply continental, with cold winters and hot summers. Soils are poor and low in carbon, nitrogen and phosphorus.

The choice of isolates for pot experiments with different plant species was justified using preliminary data on the positive effect of joint inoculations in plant nodulation assays with Baikal isolates [19,21] and the newly isolated strains of Altai origin. The results are shown in Table 2, Table 3, Table 4 and Table 5. 

In variants of co-inoculation with the commercial strain *R. leguminosarum* bv. *viciae* RCAM0626, three isolates (except the Altai isolate *Devosia* sp. A8/3-2) resulted in a significant increase in the number of nodules (Table 2). By themselves, all isolates did not form nodules or tumors on the vetch roots. An enhanced total nitrogen-fixing activity per plant was observed only when the strain *R. leguminosarum* bv. *viciae* RCAM0626 was combined with isolates *M. kowhaii* Ach-343 and *Devosia* sp. A8/3-2 (by 150 and 157%, respectively, Table 2). Co-inoculation with the last isolate also increased the level of activity calculated per one nodule compared with the commercial strain (by 143%). After co-inoculation with isolates *M. japonicum* Opo-235 and *Devosia* sp. A8/3-2, there was a significant increase in the biomass of vetch roots (by 122 and 130%, respectively, Table 3), while the isolate *M. kowhaii* Ach-343 made a smaller contribution to this parameter (116%). An enhanced total plant biomass (roots and shoots) was also observed in these three co-inoculations, but the change was not statistically different (112–114%). In general, an increase in the number of nodules did not affect the *V. sativa* biomass. The most effective was the combination of the strain *R. leguminosarum* bv. *viciae* RCAM0626 with the Altai isolate *Devosia* sp. A8/3-2, which, with almost the same number of nodules, led to an increase in nitrogen-fixing activity and plant weight.

The data of the pot experiment on mono- and co-inoculation of red clover with the commercial strain *R. leguminosarum* bv. *trifolii* RCAM1365; the isolates are shown in Table 4. Unlike common vetch plants, the isolate *M. japonicum* Opo-235 itself formed inactive nodules on clover roots. However, none of the isolates contributed to the formation of more nodules when co-inoculated with the commercial strain. In the experiment on *T. pratense*, the increase in the level of nitrogen-fixing activity per one nodule by 130–218% was detected when the commercial strain was combined with each of four isolates used (Table 4). 

This parameter was maximal in the variant *R. leguminosarum* bv. *trifolii* RCAM1365 + *P. zundukense* A18/3m. In some cases, higher activity was accompanied by an increase in the plant biomass (Table 5). Root and total plant biomass enhanced by 161 and 135%, respectively, when the strain *R. leguminosarum* bv. *trifolii* RCAM1365 was used together with the isolate *M. japonicum* Opo-242, while the shoot biomass increased by 128% when co-inoculated with the isolate *P. zundukense* A18/3m. 

Thus, the effect of joint inoculations on symbiosis with two plant species was different. On vetch, an increase in the number of nodules was mainly observed, on clover, an increase in their activity was mainly observed. This was also true for co-inoculations with the isolate *M. japonicum* Opo-235, which was used on both plants. Higher nitrogen-fixing activity did not always have a positive effect on plant biomass, which is probably due to the insufficiently long vegetation period. Nodules obtained in pot experiments were typical elongated in shape. All isolates used for joint inoculation of clover were re-isolated from nodules together with the strain *R. leguminosarum* bv. *trifolii* RCAM1365 and identified. Nodules obtained on vetch roots in the variant *R. leguminosarum* bv. *viciae* RCAM0626 + *Devosia* sp. A8/3-2 also contained both strains, while only the commercial strain was found inside the nodules in the remaining co-inoculations of *V. sativa*.

### 3.2. Whole Genome Sequences of the Studied Strains

The symbiotic and plant growth-promoting genes, as well as the genes of protein secretion systems (T3SS, T4SS and T6SS) involved in the symbiosis formation, were the target genes in all the strains used in pot experiments with common vetch and red clover. The results of the search for the symbiotic *fix*, *nif*, *nod*, *noe* and *nol* genes showed that the commercial strains and Baikal isolates *M. japonicum* Opo-235, *M. japonicum* Opo-242 and *M. kowhaii* Ach-343 [21] have common *nodABC* genes, which are usually necessary for the formation of nodules in legumes [31]. The *nif* genes (*nifHDK* and *nifENB*) as well as *fixABCX* genes, encoding components of the nitrogenase and playing a central role in electron transfer to nitrogenase [32,33], were also found in these strains. Previously, we showed that *Bradyrhizobium* sp. Opo-243 had only one cluster of *fixKJLNOQPGHIS* genes [21]. In contrast, new Altai isolates *Devosia* sp. A8/3-2, *Phyllobacterium* sp. A18/5-2 and *P. zundukense* A18/3m did not have any symbiotic clusters, but only individual genes (Table 6). Each strain had a unique set of symbiotic genes and possessed some additional *nod*, *nif*, *fix*, *nol* and *noe* genes as compared with the commercial strains. 

Presence of the *acdSR* genes, as well as genes associated with the biosynthesis of gibberellins and auxins involved in plant growth stimulation, is shown in Table 7. The commercial strains *R. leguminosarum* RCAM0626 and RCAM1365 contained no *acdS* gene encoding 1-aminocyclopropane-1-carboxylate (ACC) deaminase, which promotes nodule formation due to decreasing biosynthesis of phytohormone ethylene [34,35]. At the same time, the studied isolates had the *acdS* genes and/or *acdR* genes involved in the regulation of ACC deaminase. It can be stated that the commercial strains had significantly fewer genes associated with the biosynthesis of gibberellins and auxins compared with the isolates (Table 7). In general, Altai isolates *Devosia* sp. A8/3-2, *Phyllobacterium* sp. A18/5-2 and *P. zundukense* A18/3m were found to have the largest number of genes involved in auxin biosynthesis.

Table 8 shows the genes of T3SS, T4SS and T6SS secretion systems present in the studied strains. Among the genes of the T3SS secretion system, only the *fli* gene was found in the isolates *M. japonicum* Opo-235 and *M. kowhaii* Ach-343. The remaining isolates, like commercial strains, did not possess any genes of this secretion system. In contrast, most strains, with the exception of the *R. leguminosarum* RCAM0626, *Devosia* sp. A8/3-2 and *Phyllobacterium* sp. A18/5-2, possessed genes of the T4SS secretion system, affecting the translocation of virulence factors into the host plant cell [36,37].

Genes of the T6SS secretion system were found in only three Baikal isolates (*M. japonicum* Opo-235, *M. japonicum* Opo-242 and *M. kowhaii* Ach-343), while the last isolate had the most of them (Table 8).

The secretion of effector proteins, such as cytotoxins, lysozymes, lipoproteins, adherence factors and other compounds, due to the function of T3SS, T4SS and T6SS secretion systems, plays an important role in processes involved in bacterial virulence [38,39,40,41]. It is known that these secretion systems of rhizobia modulate nodule initiation and formation as well as the host specificity of microsymbionts [42,43,44,45].

### 3.3. Comparison of the Genomic Data and Results of Pot Experiments

To explain the role of relict isolates in increasing the efficiency of symbiosis with *V. sativa* and *T. pratense* plants, a comparative genomic analysis of the genes of strains used in pot experiments and their effects on plants was carried out. As a summarized result, the genes of rhizobial isolates related to symbiosis along with the effects of the combined inoculations with isolates and commercial strains on common vetch and red clover are presented in Table 9. We propose that at least some of these genes presented in the rhizobial isolates could be expressed and involved in the effects on symbiosis. Further discussion of the results obtained is based on this assumption. However, to confirm the proposed hypotheses, it is necessary to conduct further study of the expression of individual genes during the interaction of bacteria with plants.

Among accessory genes of the isolates that were absent in the commercial strain *R. leguminosarum* bv. *viciae* RCAM0626, the *nodPZ* and *nolK* genes participating in the modification of Nod-factors (NFs) could affect the formation of nodules and their total number on the common vetch roots. It is known that the *nodP* gene is involved in the 6-O-sulfation of the reducing end of the NFs [31], and NodP functioned in conjunction with NodQ, which synthesized the donor of the sulfate group [46]. It should be noted that the commercial strain *R. leguminosarum* bv. *viciae* RCAM0626 has the *nodQ* gene, although the importance of NFs’ sulfation for the common vetch has not been shown. The gene *nodZ* carries out fucosilation of the non-reducing end of the NFs, while the *nolK* gene is involved in the synthesis of the fucosyl group precursor from mannose and is associated with the expression of the *nodZ* gene [47,48,49]. The *nodZ* and *nolK* genes were found in various *R. leguminosarum* strains, and they are responsible for the modification of NFs, particularly the attachment of a fucosyl group that is of importance for the tribe Fabeae [50]. Isolate *Devosia* sp. A8/3-2 was the only one that increased the level of nitrogen-fixing activity per vetch nodule (i.e., not by increasing their number). It can be assumed that the symbiotic genes *nifMV* and *fixK* of the isolate *Devosia* sp. A8/3-2 could be considered as the main accessory genes contributing to the increased nitrogen fixation by the co-inoculated plants. It was previously described that the *nifM* and *nifV* genes are required, respectively, for the accumulation of active Fe protein and for the maturation of nitrogenase via homocitrate synthesis [51,52]. The coordinated functioning of *fixKL* and *fixJT* genes is necessary for optimal expression of genes regulating bacterial growth under microaerophilic conditions [53,54].

Joint inoculations of red clover increased specific nitrogen-fixing activity per nodule, which was not observed in the experiment with common vetch. The additional *nif* and *fix* genes of isolates could contribute to this parameter in the presence of *R. leguminosarum* bv. *frifolii* RCAM1365 (Table 9). For example, genes *nifZQXW* are involved in the synthesis of the Fe-S-cofactor and fixation of molybdenum, protecting nitrogenase from the negative effects of oxygen [51,55,56,57,58]. The *fixHSQ* genes play important role in the biosynthesis of the oxidase complex, which regulates the transport of oxygen through membranes, allowing normal cell breath at low oxygen concentrations [53,56]. The role in the symbiosis of the *fixKL* genes, as well as the gene *nifV*, is described above. In all variants of joint inoculation with an increased level of nitrogen-fixing activity, both strains were present in nodules. In our previous work, the isolates *M. japonicum* Opo-235 and Opo-242 also increased the nitrogen-fixing activity per one nodule when co-inoculated with the strain *R. leguminosarum* bv. *trifolii* RCAM1365 in the plant nodulation assay [21]. Another study showed an increase in the nodule activity on *Glycyrrhiza uralensis* plants co-inoculated with the *M. japonicum* Opo-235 and *M. kowhaii* Ach-343 strains carrying the complementary *nifQV* and *fixJKL* genes. The important observation was that these isolates, having genetic complementation to each other, were localized in the same plant cells [19].

It should be noted that in addition to symbiotic genes, the genes related to plant growth promotion (*acdSR*, gibberellin- and auxin-biosynthesis related genes), as well as the genes of the T3SS, T4SS and T6SS secretion systems, can also positively influence the process of nodulation and nitrogen-fixing activity (Table 9). It was shown that these genes involved in the biosynthesis of phytohormones and key components of bacterial pathogenicity (e.g., penetrating effectors and cytotoxins) can participate in the integration between symbiotic partners [39,40,41,59,60,61]. Further study of the phenomenon of synergistic rhizobial interaction based on the complementarity of genes in taxonomically different co-microsymbionts will help reveal the mechanisms for the formation of effective integration of partners and the evolution of legume–rhizobium symbiosis.

Thus, this study and the results obtained earlier show that strains isolated from relict legumes have markedly more genes that affect plant–microbial interactions than commercial strains isolated from currently cultivated crops. At the same time, relict isolates can occupy nodules formed by commercial strains, even if they cannot form them on their own. Since the level of cooperation between different co-microsymbionts most likely depends on their spatial arrangement (rhizosphere, rhizoplane or nodule), the localization of rhizobia in the same nodule (or even in the same plant cell) should lead to an increase in the level of their integration. To implement such integration, a necessary condition is the expression of the corresponding genes in relict isolates and commercial strains during co-inoculation. A detailed study of this issue will make it possible to understand the mechanisms of the observed synergy. At this stage, we can only assume that the observed effects may be associated with the modification of nod-factors under the influence of relict isolates and horizontal gene transfer between microsymbionts. This cooperation between different rhizobia, including relict isolates and commercial strains, can be expressed with a positive change in symbiotic parameters (number of nodules, nitrogen-fixing activity and plant biomass) during joint inoculation. 

It can be assumed that this study is not only of applied importance for agricultural biotechnology, but is also of great fundamental interest. In particular, based on a comparison of the whole genome analysis of strains and their effect on plants, preliminary conclusions can be drawn about the role of different genes in the formation of effective symbiosis.

## Figures and Tables

**Table 1 microorganisms-11-01206-t001:** Commercial strains and rhizobial isolates from the relict legumes of the Baikal Lake region and the Altai Republic used in the pot experiment.

Origin of Russia	Host Plant	Strain/Isolate	Species of Rhizobia
Commercial strain			
Leningrad region	*Vicia sativa*	RCAM0626	*Rhizobium leguminosarum* bv. *viciae*
Moscow region	*Trifolium pratense*	RCAM1365	*R. leguminosarum* bv. *trifolii*
Isolate			
Baikal Lake region	*Oxytropis* *popoviana*	Opo-235	*Mesorhizobium japonicum*
		Opo-242	*M. japonicum*
		Opo-243	*Bradyrhizobium* sp.
	*Astragalus horinensis*	Ach-343	*M. kowhaii*
Altai Republic	*O. tragacanthoides*	A8/3-2	*Devosia* sp.
	*V. costata*	A18/5-2	*Phyllobacterium* sp.
		A18/3m	*P. zundukense*

**Table 2 microorganisms-11-01206-t002:** Number of nodules and acetylene reduction activity after inoculation of *V. sativa* plants with the commercial strain *R. leguminosarum* bv. *viciae* RCAM0626 and the isolates *M. japonicum* Opo-235, *Bradyrhizobium* sp. Opo-243, *M. kowhaii* Ach-343 and *Devosia* sp. A8/3-2 in the pot experiment. Data are means ± standard errors of one representative experiment (*n* = 7). Asterisks show significant difference against the treatment with the strain RCAM0626 (Fisher’s LSD test, * *p* < 0.05, ** *p* < 0.01, *** *p* < 0.001). AV—absolute value, RV—relative value to the strain RCAM0626.

Treatment	Number of Nodules (pot^−1^)		Acetylene Reduction Activity			
			µmol C_2_H_4_ plant^−1^ h^−1^		nmol C_2_H_4_ nodule^−1^ h^−1^	
	AV	RV, %	AV	RV, %	AV	RV, %
Without inoculation	0	0	0	0	0	0
**RCAM0626**	**92 ± 5**	**100**	**1.4 ± 0.2**	**100**	**30 ± 4**	**100**
RCAM0626 + Opo-235	113 ± 7 **	123	1.5 ± 0.1	107	27 ± 2	90
RCAM0626 + Opo-243	110 ± 7 *	120	1.7 ± 0.2	121	32 ± 3	107
RCAM0626 + Ach-343	132 ± 8 ***	143	2.1 ± 0.2 **	150	32 ± 3	107
RCAM0626 + A8/3-2	103 ± 6	112	2.2 ± 0.2 ***	157	43 ± 4 **	143
Opo-235	0	0	0	0	0	0
Opo-243	0	0	0	0	0	0
Ach-343	0	0	0	0	0	0
A8/3-2	0	0	0	0	0	0

**Table 3 microorganisms-11-01206-t003:** Biomass of *V. sativa* plants after inoculation with the commercial strain *R. leguminosarum* bv. *viciae* RCAM0626 and the isolates *M. japonicum* Opo-235, *Bradyrhizobium* sp. Opo-243, *M. kowhaii* Ach-343 and *Devosia* sp. A8/3-2 in the pot experiment. Data are means ± standard errors of one representative experiment (*n* = 14). Asterisks show significant difference against the treatment with the strain RCAM0626 (Fisher’s LSD test, * *p* < 0.05, ** *p* < 0.01, *** *p* < 0.001). Plant, root and shoot biomass in mg fresh weight plant^−1^, AV—absolute value, RV—relative value to the strain RCAM0626.

Treatment	Plant Biomass		Root Biomass		Shoot Biomass	
	AV	RV, %	AV	RV, %	AV	RV, %
Without inoculation	1663 ± 89 ***	43	1204 ± 60 **	72	459 ± 39 ***	21
**RCAM0626**	**3845 ± 251**	**100**	**1663 ± 131**	**100**	**2183 ± 149**	**100**
RCAM0626 + Opo-235	4310 ± 292	112	2030 ± 167 *	122	2280 ± 147	104
RCAM0626 + Opo-243	4065 ± 306	106	1782 ± 166	107	2283 ± 155	105
RCAM0626 + Ach-343	4345 ± 195	113	1925 ± 123	116	2420 ± 127	111
RCAM0626 + A8/3-2	4387 ± 169	114	2161 ± 97 **	130	2226 ± 101	102
Opo-235	1560 ± 78 ***	41	1031 ± 51 ***	62	529 ± 35 ***	24
Opo-243	2283 ± 156 ***	59	1650 ± 133	99	633 ± 56 ***	29
Ach-343	2295 ± 131 ***	60	1742 ± 101	105	553 ± 41 ***	25
A8/3-2	1530 ± 116 ***	40	1006 ± 86 ***	60	524 ± 52 ***	24

**Table 4 microorganisms-11-01206-t004:** Number of nodules and acetylene reduction activity after inoculation of *T. pratense* plants with the commercial strain *R. leguminosarum* bv. *trifolii* RCAM1365 and the isolates *M. japonicum* Opo-235 and Opo-242, *Phyllobacterium* sp. A18/5-2 and *P. zundukense* A18/3m in the pot experiment. Data are means ± standard errors of one representative experiment (*n* = 7). Asterisks show significant difference against the treatment with the strain RCAM1365 (Fisher’s LSD test, * *p* < 0.05, *** *p* < 0.001). AV—absolute value, RV—relative value to the strain RCAM1365.

Treatment	Number of Nodules (pot^−1^)		Acetylene Reduction Activity			
			µmol C_2_H_4_ plant^−1^ h^−1^		nmol C_2_H_4_ nodule^−1^ h^−1^	
	AV	RV, %	AV	RV, %	AV	RV, %
Without inoculation	0	0	0	0	0	0
**RCAM1365**	**22 ± 2**	**100**	**3.2 ± 0.2**	**100**	**295 ± 19**	**100**
RCAM1365 + Opo-235	20 ± 3	91	3.8 ± 0.3	119	383 ± 27 *	130
RCAM1365 + Opo-242	24 ± 2	109	5.8 ± 0.4 ***	181	489 ± 33 ***	166
RCAM1365 + A18/5-2	26 ± 3	118	5.1 ± 0.3 ***	159	390 ± 26 *	132
RCAM1365 + A18/3m	17 ± 2	77	5.5 ± 0.4 ***	172	645 ± 50 ***	218
Opo-235	25 ± 2	114	0	0	0	0
Opo-242	0	0	0	0	0	0
A18/5-2	0	0	0	0	0	0
A18/3m	0	0	0	0	0	0

**Table 5 microorganisms-11-01206-t005:** Biomass of *T. pratense* plants after inoculation with the commercial strain *R. leguminosarum* bv. *trifolii* RCAM1365 and the isolates *M. japonicum* Opo-235 and Opo-242, *Phyllobacterium* sp. A18/5-2 and *P. zundukense* A18/3m in the pot experiment. Data are means ± standard errors of one representative experiment (*n* = 14). Asterisks show significant difference against the treatment with the strain RCAM1365 (Fisher’s LSD test, ** *p* < 0.01, *** *p* < 0.001). Plant, root and shoot biomass in mg fresh weight plant^−1^, AV—absolute value, RV—relative value to the strain RCAM1365.

Treatment	Plant Biomass		Root Biomass		Shoot Biomass	
	AV	RV, %	AV	RV, %	AV	RV, %
Without inoculation	170 ± 12 ***	45	98 ± 8 ***	63	72 ± 5 ***	33
**RCAM1365**	**376 ± 26**	**100**	**155 ± 13**	**100**	**221 ± 17**	**100**
RCAM1365 + Opo-235	323 ± 24	86	111 ± 6 **	72	213 ± 23	96
RCAM1365 + Opo-242	506 ± 33 ***	135	250 ± 15 ***	161	256 ± 26	116
RCAM1365 + A18/5-2	405 ± 32	108	151 ± 16	97	254 ± 25	115
RCAM1365 + A18/3m	421 ± 32	112	138 ± 12	89	282 ± 25 **	128
Opo-235	167 ± 13 ***	44	101 ± 8 ***	65	67 ± 6 ***	30
Opo-242	174 ± 12 ***	46	113 ± 10 **	73	61 ± 3 ***	28
A18/5-2	202 ± 14 ***	54	135 ± 9	87	67 ± 6 ***	30
A18/3m	196 ± 12 ***	52	123 ± 9 **	79	73 ± 5 ***	33

**Table 6 microorganisms-11-01206-t006:** Presence of the symbiotic genes *fix*, *nif*, *nod*, *noe* and *nol* in the commercial rhizobial strains *R. leguminosarum* bv. *viciae* RCAM0626 and *R. leguminosarum* bv. *trifolii* RCAM1365 used in the pot experiments, as well as three strains isolated from the Altai relict legumes *O. tragacanthoides* (*Devosia* sp. A8/3-2) and *V. costata* (*Phyllobacterium* sp. A18/5-2 and *P. zundukense* A18/3m).

Genes	Isolates from the Altai Relict Legumes			Commercial Strains	
	A8/3-2	A18/5-2	A18/3m	RCAM0626	RCAM1365
Fix copy 1	−	−	−	NOQP	−
Fix copy 2	−	−	−	LNOQPGHI	−
Fix cluster	−	−	−	ABCXLNOQPGHI	GINOPABCX
*fixBA*	+	+	+	+	+
*fixC*	+	+	+	−	−
*fixI*	+	+	+	−	−
*fixJ*	+	+	+	+	+
*fixK*	+	+	+	−	−
*fixL*	+	+	+	−	−
*fixN*	+	−	−	−	−
*fixP*	+	−	−	-	−
Nif cluster	−	−	−	HDKENAB	HDKENABT
*nifA*	+	+	+	−	−
*nifL*	+	+	+	+	+
*nifM*	+	−	−	−	−
*nifR*	+	-	-	+	−
*nifS*	+	+	+	+	+
*nifU*	+	+	+	+	+
*nifV*	+	+	+	−	−
Nod cluster	−	−	−	JICBADEFLMNO	JICBADEFLMND
*nodG*	+	+	+	+	+
*nodM*	+	+	+	+	−
*nodN*	−	+	+	+	+
*nodP*	+	+	+	−	+
*nodQ*	+	+	+	+	+
*nodT*	−	−	−	+	+
*nodV*	-	+	-	−	−
*nodW*	-	+	+	+	+
*noeK*	+	+	-	+	+
*noeL*	-	+	+	+	+
*nolG*	+	−	−	−	−
*nolK*	−	+	+	-	-
*nolF*	−	−	−	+	+
*nolL*	+	+	−	+	+
*nolR*	−	−	−	−	+

**Table 7 microorganisms-11-01206-t007:** Presence of the ACC deaminase genes, as well as genes associated with the biosynthesis of auxins and gibberellins in the commercial rhizobial strains *R. leguminosarum* bv. *viciae* RCAM0626, *R. leguminosarum* bv. *trifolii* RCAM1365 and seven strains isolated from the relict legumes. Genes only present in isolates are marked in pink.

Features/Genes	Isolates from the Relict Legumes							Strain RCAM	
	Opo- 235	Opo- 242	Opo- 243	Ach- 343	A8/3-2	A18/5-2	A18/3m	0626	1365
ACC deaminase synthesis									
*acdS*	+	+	+	+	−	−	−	−	−
*acdR*	-	-	+	+	+	+	+	−	−
Auxin synthesis									
amine oxidase	+	+	+	+	−	−	−	−	−
indole synthase	−	−	−	−	+	+	+	−	−
indole-3-acetaldehyde dehydrogenase (*aldA*)	−	−	−	−	+	+	+	−	−
indole-3-acetamide (IAM) hydrolase (*iaaH*)	−	−	+	+	+	+	+	−	−
tryptophan-2-monooxygenase (*iaaM*)	−	−	−	−	−	+	+	−	−
indole-3-pyruvate decarboxylase (*ipdC*)	−	−	−	−	+	+	+	−	−
nitrile hydratase subunits alpha, beta	+	+	+	+	-	+	+	+	+
tryptophan decarboxylase (TDC)	−	−	−	−	+	+	+	−	−
tryptophan synthase (*trp*)	+	+	+	+	+	+	+	-	-
Gibberellin synthesis									
*cpxP* (CYP112)	+	+	+	+	+	+	+	−	−
*cpxR* (CYP114)	+	+	+	+	+	+	+	−	−
*cpxU* (CYP117)	+	+	+	+	+	+	−	−	−
ferredoxin (FD)	−	−	−	−	+	+	+	−	−
*ispA*	+	+	+	+	+	+	+	−	−
SDR family	+	+	+	−	+	+	+	−	−
Cluster CPS (copalyl diphosphate synthase)	−	−	−	+	−	−	−	−	−

**Table 8 microorganisms-11-01206-t008:** Presence of the secretion systems T3SS, T4SS and T6SS genes in the commercial rhizobial strains *R. leguminosarum* bv. *viciae* RCAM0626, *R. leguminosarum* bv. *trifolii* RCAM1365 and seven strains isolated from the relict legumes.

Bacterial Species	Strain	Secretion System		
		T3SS	T4SS	T6SS
*R. leguminosarum*	RCAM0626	−	−	−
	RCAM1365	-	*virB1-6,8-11*	−
*M. japonicum*	Opo-235	*fli*	*virB2-6,8-11*	*icmF*, *tssABCEGJKL*, *tagFH*, *vgrG*, *vasA*, *clpV*, Hcp family
*M. japonicum*	Opo-242	−	*virB1-6,8-11*	*clpV1*, *tssABCDEFG*, *tagH*, *tssJKLM*, *tagF*, PAAR domain, Vgr family, *tagH*, *tssLM*
*Bradyrhizobium* sp.	Opo-243	−	*virB1-6,8-11*	−
*M. kowhaii*	Ach-343	*fli*	*virB2-11*	*icmF*, *tssABCEGJKL*, *tagFH*, *vgrG*, *vasAEFK*, *clpV1*, *impA*, *vipAB*, *sciN*, *dotU*, Vgr family, Hcp family, FHA domain
*Devosia* sp.	A8/3-2	−	−	−
*Phyllobacterium* sp.	A18/5-2	−	−	−
*P. zundukense*	A18/3m	−	*virB1-11*	−

**Table 9 microorganisms-11-01206-t009:** Symbiotic and non-symbiotic genes present in the isolates but not in the commercial strains of common vetch and red clover and the effects of co-inoculations. SG stands for symbiotic genes, NSG—non-symbiotic genes. Effect: NN, PB, RB and SB—increased number of nodules, plant biomass, root biomass and shoot biomass per plant, respectively; ARAp and ARAn—increased acetylene reduction activity per plant and per nodule, respectively.

Isolate	Commercial Strain						
	*R. leguminosarum* RCAM0626				*R. leguminosarum* RCAM1365		
	SG	NSG	Effect		SG	NSG	Effect
*M. japonicum* Opo-235	*nodPZ*, *nifTVWXZ*, *fixKS*	T3SS, T4SS, T6SS, *acdS*, *trp*, amine oxidase, *cpxPRU*, *ispA*, SDR family	NN, RB		*nodZ*, *nifVWXZ*, *fixHKLQS*	T3SS, T6SS, *acdS*, *trp*, amine oxidase, *cpxPRU*, *ispA*, SDR family	ARAn, RB
*M. japonicum* Opo-242	Not detected				*nodZ*, *nifQWXZ*, *fixHKLQS*	T6SS, *acdS*, *trp*, amine oxidase, *cpxPRU*, *ispA*, SDR family	ARApn, PB, RB
*Bradyrhizobium* sp. Opo-243	*nodP*, *nolK*, *fixKS*	T4SS, *acdRS*, *iaaH*, *trp*, amine oxidase, *cpxPRU*, *ispA*, SDR family		NN	Not detected		
*M. kowhaii* Ach-343	*nodPZ*, *nifQTWXZ*, *fixS*	T3SS, T4SS, T6SS, *acdRS*, *iaaH*, *trp*, amine oxidase, *cpxPRU*, *ispA*, CPS cluster		NN, ARAp			
*Devosia* sp. A8/3-2	*nodP*, *nolG*, *nifMV*, *fixK*	*acdR*, indole synthase, *aldA*, *iaaH*, *ipdC*, TDC, *trp*, *cpxPRU*, FD, *ispA*, SDR family		ARApn, RB			
*Phyllobacterium* sp. A18/5-2	Not detected				*nodV*, *nolK*, *nifV*, *fixKL*	*acdR*, indole synthase, *aldA*, *iaaH*, *iaaM*, *ipdC*, TDC, *trp*, *cpxPRU*, FD, *ispA*, SDR family, T4SS	ARApn
*P. zundukense* A18/3m					*nolK*, *nifV*, *fixKL*	*acdR*, indole synthase, *aldA*, *iaaH*, *iaaM*, *ipdC*, TDC, *trp*, *cpxPR*, FD, *ispA*, SDR family, T4SS	ARApn, SB

## Data Availability

Not applicable.

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
