# Peer review of "Synergy between Rhizobial Co-Microsymbionts Leads to an Increase in the Efficiency of Plant–Microbe Interactions"

_microorganisms, 2023, doi:10.3390/microorganisms11051206_

Round 1

Reviewer 1 Report

The article is interesting, relevant and well-written.His research to expand knowledge about the synergistic effects between commercial rhizobia of pasture legumes and root nodule bacteria of relict legume species. It is interesting research, and much effort has been put in. Paper is well written and shows interesting results and data. The conclusions are consistent with the evidence and arguments presented. The article is original and relevant in its field. In connection with all of the above, I recommend accepting this article for publication.

Author Response

Thans for your positive review

Reviewer 2 Report

Lines 241-316: point 2.3. To me this point should be the Discussion section, and, in my opinion, it should be rewritten for several reasons. 1) The fact that a bacterial strains harbour certain genes does not necessarily mean that these genes are expressed during the interaction with plants. 2) Automatic annotation of bacterial genomes sometimes may lead to confusions and errors: for example, the “nod genes” present in the isolates of the Altai relict legumes (Table 6) are actually real nod genes or, instead, they code for protein with similar functions to the corresponding Nod proteins? Could the level of identity to known nod genes be indicated in the Table? 3) Are you claiming that the putative Nod proteins of “the isolates” might change the decoration of the Nod factors produced by the commercial strains (lines 247-259), or that Nif/Fix proteins of those “isolates” might affect nitrogenase activity in the commercial strain (lines 260-268)? To me it is difficult conceiving how such phenomena might take place: Nod factors passing form one bacterium to another? Lateral transfer of genes? This should be further explained. Indeed, did you check whether the presence of “isolates” change the composition of the Nod factors produced by the “commercial strains”. 

Round 2

Reviewer 2 Report

Although the manuscript has clearly improved, there are still some concerns that the authors have not addressed:

1)  Line 45: rhizobia do not form nodules but induce its formation by the plant. Please, correct here and elsewhere in the whole article (did not have the ability to induce their formation….).

2)  Line 118: “data not published” is not acceptable as a result. Please, provide the necessary information as supplementary data.

3)      Line 187: “ harbour the nodABC genes, which are indispensable for legume nodulation in most cases”. As you know, there are some cases in which T3SS can promote nodulation in the absence of Nod factors (please, check the three reviews I mention below)

4)      Line 237: reference 33. The relevance of rhizobial T3SS has been thoroughly studied in the past few years. Maybe you can add a more recent review as a reference, such as

                Ratu STN, Amelia L, Okazaki S. Type III effector provides a novel symbiotic pathway in legume-rhizobia symbiosis. Biosci Biotechnol Biochem. 2022 Dec 21;87(1):28-37. doi: 10.1093/bbb/zbac178. PMID: 36367542.

                Jiménez-Guerrero I, Medina C, Vinardell JM, Ollero FJ, López-Baena FJ. The Rhizobial Type 3 Secretion System: The Dr. Jekyll and Mr. Hyde in the Rhizobium-Legume Symbiosis. Int J Mol Sci. 2022 Sep 21;23(19):11089. doi: 10.3390/ijms231911089. PMID: 36232385; PMCID: PMC9569860.

                Teulet A, Camuel A, Perret X, Giraud E. The Versatile Roles of Type III Secretion Systems in Rhizobium-Legume Symbioses. Annu Rev Microbiol. 2022 Sep 8;76:45-65. doi: 10.1146/annurev-micro-041020-032624. Epub 2022 Apr 8. PMID: 35395168.
